# The Onset of Subtalar Joint Monoarthritis in a Patient with Rheumatoid Arthritis

**DOI:** 10.3390/diagnostics12102311

**Published:** 2022-09-25

**Authors:** Hiroki Wakabayashi, Kenta Nakata, Akinobu Nishimura, Masahiro Hasegawa, Akihiro Sudo

**Affiliations:** Department of Orthopedic Surgery, Mie University Graduate School of Medicine, Tsu 514-8507, Japan

**Keywords:** subtalar joint, monoarthritis, rheumatoid arthritis

## Abstract

The involvement of the subtalar joint is uncommon in the early stages of rheumatoid arthritis (RA). We report a case of a 47-year-old female who had RA with isolated subtalar joint arthritis. The clinical history, magnetic resonance imaging, and pathological findings of the patient are presented. A careful evaluation of the patients for chronic ankle-to-heel pain should be conducted, and concomitant evaluation for inflammatory arthritis, including RA, should be considered.

## 1. Introduction

Foot involvement is not uncommon in rheumatoid arthritis (RA). The frequency of foot deformities in RA patients is reportedly 78%, and metatarsus primus varus account for 62% and the splaying of the forefoot for 41% of these types of deformities [1]. Changes in the hindfoot tend to occur later in the disease course than those of the forefoot [2]. The subtalar joint is rarely affected in the early stages of rheumatic pathogenesis. Therefore, heel pain is frequently interpreted as the result of overuse or trauma. Isolated subtalar joint arthritis is rarely described as the sole manifestation of RA. Herein, we present a case of RA with chronic subtalar joint monoarthritis.

## 2. Case Presentation

A 47-year-old female presented to her orthopedist with a 2-year history of ankle-to-heel pain. She did not have a history of trauma and smoking or a family history of rheumatic diseases. Examination demonstrated tenderness only over the right subtalar joint. Although her plain radiographs were normal (Figure 1a,b), in non-contrast-enhanced magnetic resonance imaging (MRI), the T1-weighted image showed a relatively low-isointense signal, and the T2-weighted image showed a relatively hyperintense signal, compared with those for the skeletal muscle in the subtalar joint and the surrounding bone (Figure 1c–f). These images appeared to reflect moderate subtalar joint effusion with bony edema and surrounding synovitis.

Moreover, MRI showed an effusion of the tibiotalar joint, the posterior tibial tendon, and the flexor hallucis longus tendon, suggesting tendonitis. The patient refused a biopsy because treatment with non-steroidal anti-inflammatory drugs improved her ankle-to-heel pain. Therefore, the patient was examined at an outpatient clinic with orthopedic tests, including plain radiography and MRI.

One year later, a follow-up MRI showed a remarkable change in synovial enhancement surrounding the subtalar joint. The bone destruction of the subtalar joint had progressed, and inflammatory arthritis had spread to the tibiotalar joint (Figure 2a–d). Thus, computed tomography (CT) was performed, which revealed the erosion of the subtalar joint (Figure 3a–c). After MRI and CT, the patient presented for a rheumatologic evaluation. Her white blood cell count (4880/μL), erythrocyte sedimentation rate (9 mm/h), C-reactive protein levels (0.12 mg/dL), and anti-nuclear antibody levels were normal. However, anti-cyclic citrullinated peptide antibody (83.7 U/mL), rheumatoid factor (IgM-RF 22 U/mL), and matrix metalloproteinase-3 (83.7 ng/ml) levels were elevated. Her ankle-to-heel pain gradually worsened.

Arthroscopic biopsy and synovectomy were performed. As the bone destruction of the subtalar joint had already progressed, subtalar joint fusion was performed with synovectomy. Intra-operatively, the subtalar joint was found to be stable; however, the cartilage had degenerated, and the synovium was severely inflamed. The histological analysis of the biopsy specimen showed synovial proliferation and lymphocytes. In the synovium, the synovial lining cell and vascular proliferation and lymphocyte infiltration were observed (Figure 4a,b). Subtalar joint arthritis subsequently resolved after synovectomy and joint fusion. Although the patient had monoarthritis, her serological test results (rheumatoid factor and anti-cyclic citrullinated peptide antibody) were positive without a smoking history. She did not fulfill the criteria of the 2010 American College of Rheumatology/European Alliance of Associations for Rheumatology classification for RA; dactylitis, spondyloarthritis, or extra-articular manifestations such as uveitis, psoriasis, or inflammatory bowel disease were not present during the evaluation. She was diagnosed with RA based on imaging and pathological findings.

The plain radiographs of her hands and left foot were normal; significant atrophy was seen in the right foot compared with the left foot (Appendix A). After surgery, she continued treatment with non-steroidal anti-inflammatory drugs but occasionally experienced proximal interphalangeal joint pain in her fingers, but no swelling was present. Therefore, RA treatment with methotrexate was started 3 months after surgery. Nine months after surgery, the subtalar joint fusion was complete. Her disease activity remained in remission (tender joints, 0; swollen joints, 0; C-reactive protein, 0.07 mg/dL; patient’s global assessment, 0 mm; physician’s global assessment, 0 mm) according to the simplified disease activity index.

## 3. Discussion

Ongoing joint inflammation with synovitis leads to joint destruction and deformation, as well as chronic pain and disability. In the early stage of the disease, the most commonly affected areas are small joints such as the wrists, metacarpophalangeal joints, proximal interphalangeal joints of fingers, and metatarsophalangeal joints. However, with disease progression, large joints such as shoulders, elbows, knees, feet, and ankles may also become involved if the diagnosis is delayed, and early treatment is not provided [3,4]. The subtalar joint is less frequently affected by RA than other synovial joints [2,5]; however, its progressive damage can lead to a flat foot through a synergistic effect that involves the weakening of the capsule, ligament, and exaggerated pronation forces [2].

Șerban et al. found that the quality of life (QoL) of patients with RA is significantly affected due to ankle and hindfoot pathology, including tibiotalar synovitis, subtalar synovitis, tibialis posterior tenosynovitis/tendinosis, and peroneal tendinosis. Inflammation, as well as degeneration and deformities, were diagnosed [6]. Tibialis posterior tendon involvement on MRI was frequently observed in those patients with RA presenting with tarsal inflammation [7]. In our case, the initial MRI indicated subtalar and tibiotalar arthritis, and the subtalar joint showed a change in intensity and bone marrow edema. In addition, tendonitis was observed in the right posterior tibial tendon and flexor hallucis longus tendon.

Foot pain is common in RA, from the forefoot involvement in early RA to the later involvement of the hindfoot (talonavicular and subtalar joints). Spiegel et al. observed that only 8% of patients with rheumatoid disease for less than 5 years demonstrated hindfoot changes, whereas these changes were present in 25% of patients with the disease for over 5 years.

Foot and ankle involvement can lead to reduced walking distance and affect QoL [8]. In addition, the involvement of the hindfoot is associated with functional disability and walking difficulties [9]. In recent reports, approximately half the RA patients report foot or ankle joint symptoms as the first manifestation of the disease [10,11], and 71% of these patients develop walking disabilities over time [12]. Additionally, the radiographical assessment of the ankle and subtalar joints in patients with early RA within 6 months revealed that subtalar joint damage appeared within the first year of follow-up [5]. In our case, from the initial MRI and the follow-up MRI, we determined that the inflammation spread to the tibiotalar joint after initially developing in the subtalar joint due to bone marrow edema. This case demonstrates the onset of damage to the subtalar joint in RA.

Preceding subtalar destruction and alignment should be carefully monitored and treated to avoid excessive stress on the ankle joint in patients with RA. Subtalar joint pain is mainly expressed in the lateral hindfoot during activity due to chronic inflammation and destruction. If untreated, progressive eversion at the subtalar joint, together with the dysfunction of peritalar ligaments and the tibialis posterior tendon, subsequently leads to the instability of the subtalar and midtarsal joints [13,14]. However, hindfoot correction can potentially reduce ankle joint pain through the realignment of the loading axis [15].

Monoarticular RA is rare, and reaching a diagnosis can be challenging. However, a case series involved four patients diagnosed with chronic monoarthritis (three of the knee and one involving the metacarpophalangeal joint). Further workup such as histopathological and radiological analyses (MRI and ultrasound) revealed a diagnosis of seropositive RA [16].

RA treatment usually starts with methotrexate. In this case, the bone destruction of the subtalar joint had already progressed; therefore, histological workup, followed by synovectomy and joint fusion surgery, was performed first. To minimize surgical invasion and reduce the risk of complications for subtalar arthrodesis, we proposed an arthroscopic posterior approach to implant an osseous allograft into the joint space. As its advantages include the minimal invasion of the soft tissue around the hindfoot and the preservation of the blood supply to the talus, arthroscopic subtalar arthrodesis has been reported as an alternative to traditional open methods for intractable hindfoot disorders such as subtalar arthritis after fracture and inflammatory arthritis [17]. In the present case, the subtalar joint fusion was completed without complications; however, careful observation is required because the tibiotalar joint space remained narrow and because of the future possibility of polyarthritis.

We reported a case of subtalar joint monoarthritis in a patient with RA. Inflammatory arthritis of the subtalar joint is uncommon and underdiagnosed because of ankle-to-heel pain. RA, in this case, progressed due to delayed diagnosis, and arthroscopic subtalar arthrodesis was performed. After surgery for monoarthritis of the subtalar joint, RA was diagnosed based on radiological findings, the synovial histological analysis, the results of serological tests, and the occasional complaint of small joint pain. This novel case highlights the need for careful evaluation of inflammatory arthritis, including RA, in patients with chronic ankle-to-heel pain.

## Figures and Tables

**Figure 1 diagnostics-12-02311-f001:**
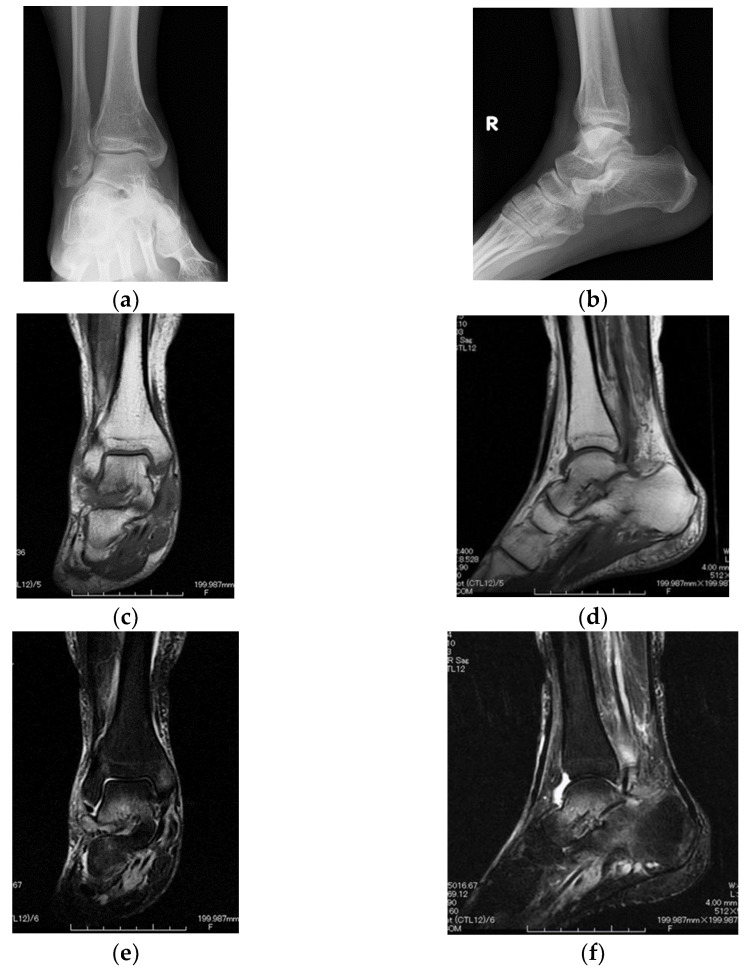
Subtalar joint involvement in a 47-year-old female. Anteroposterior (**a**) and lateral (**b**) plain radiographs of the right ankle joint. Coronal T1-weighted (**c**), sagittal T1-weighted (**d**), coronal fat-suppressed T2-weighted (**e**), and sagittal fat-suppressed T2-weighted (**f**) images of the right ankle joint using magnetic resonance imaging (MRI). MRI images show a moderate subtalar joint effusion with bone marrow edema.

**Figure 2 diagnostics-12-02311-f002:**
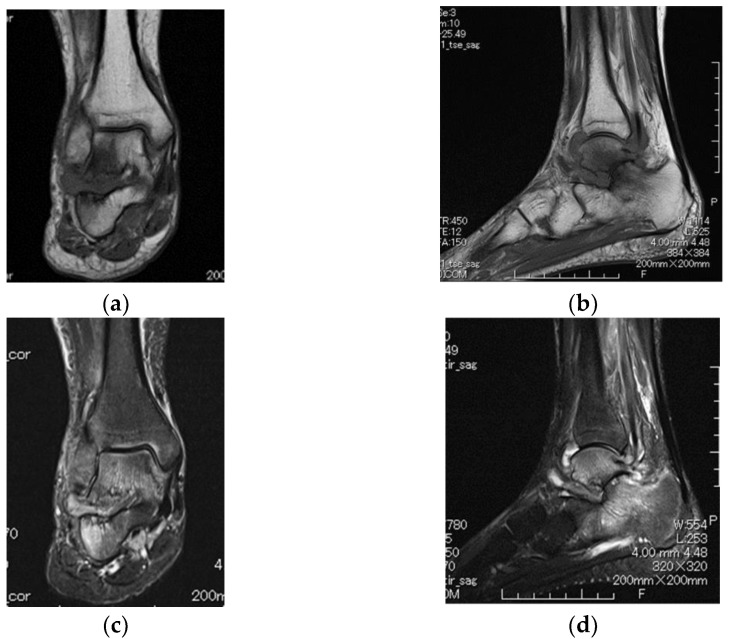
Subtalar joint showing progressive involvement 1 year later. Coronal T1-weighted (**a**), sagittal T1-weighted (**b**), coronal fat-suppressed T2-weighted (**c**), and sagittal fat-suppressed T2-weighted (**d**) images of the right ankle joint. Findings show a marked loss of articular cartilage with associated bone marrow edema in the subarticular region of the subtalar joint and bone marrow edema of the tibiotalar joint.

**Figure 3 diagnostics-12-02311-f003:**
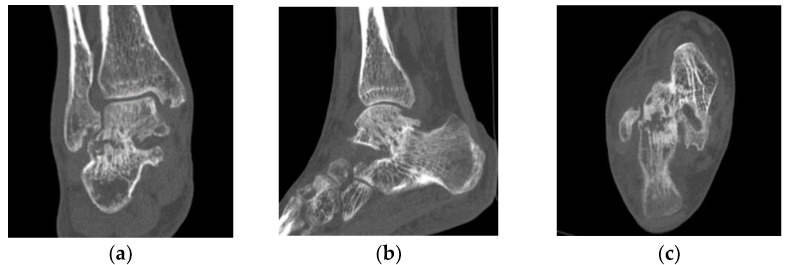
Erosion and destruction of the subtalar joint on computed tomography. Computed tomography images of the coronal (**a**), sagittal (**b**), and axial (**c**) parts of the subtalar joint. Findings demonstrate narrowed joint space and destruction of the subtalar joint of the right ankle.

**Figure 4 diagnostics-12-02311-f004:**
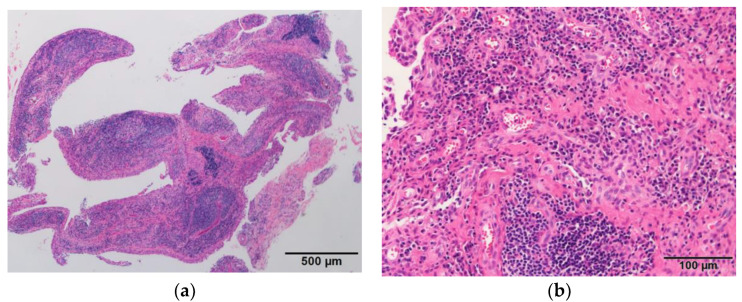
Surgical biopsy specimen of the subtalar joint region shows synovial proliferation and lymphocytes in the synovium (**a**). Proliferation of synovial lining cells, vascular proliferation, and lymphocyte infiltrate can be observed in the synovium (**b**).

## Data Availability

No new data were created or analyzed in this study. Data sharing is not applicable to this article.

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
