# Peer review of "The Onset of Subtalar Joint Monoarthritis in a Patient with Rheumatoid Arthritis"

_diagnostics, 2022, doi:10.3390/diagnostics12102311_

Round 1

Reviewer 1 Report

The case is interesting.

Monoarticular rheumatoid arthritis is rare and challenging. I would recommend briefly reviewing and discussing other cases presenting as monoarticular rheumatoid arthritis in the discussion including clinical presentation, most frequently affected joints, treatment, and prognosis. This article may help 10.5152/eurjrheum.2017.17011.

Please review the English spelling errors

Example: line 19 “matarsus primus”

She is a patient with chronic monoarthritis with positive ACPA and weakly positive RF but she did not fulfill the 2010 ACR/EULAR classification criteria of RA. Although the patient may had had RA, other causes of chronic monoarthritis are not ruled out, including spondyloarthropathies, which are more likely to affect the foot and cause chronic monoarthritis. Nor is it mentioned whether the patient smoked, since smoking can be associated with positive ACPA.

I agree with the published information by Yano K., et al, (2018) that the ankle may be an initial manifestation. But, I consider that all these patients eventually will progress to polyarthritis to make a definite diagnosis of RA. This patient had a DAS 28 that places her in remission since 28-joint count assessments do not include examination of the foot and ankle. Did you order joint ultrasounds of MCP and wrist joints?

Although, this study is not included in the 2010 ACR/EULAR criteria either.

The patient requires a follow-up to know if she develops defined rheumatoid arthritis in the future.

Author Response

Replies to Reviewer #1

On the comment of [The case is interesting.]

Authors are grateful to Reviewer #1 for encouraging comments. We have revised the indicated parts of the manuscript according to the comments. Corrections in the newly revised manuscript are underlined.

Please note that your review comments are shown in italic below and our replies in non-italic.

On the comment of [Monoarticular rheumatoid arthritis is rare and challenging. I would recommend briefly reviewing and discussing other cases presenting as monoarticular rheumatoid arthritis in the discussion including clinical presentation, most frequently affected joints, treatment, and prognosis. This article may help 10.5152/eurjrheum.2017.17011.]

Reply: Thank you for suitable suggestion. We refereed the 10.5152/eurjrheum.2017.17011 as ref.16, and added discuss on line136-139.

On the comment of [Please review the English spelling errors. Example: line 19 “matarsus primus”]

Reply: Thank you for suitable suggestion. We reviewed the English spelling errors and corrected ”metatarsus primus”. And we have reviewed the English language again.

On the comment of [' She is a patient with chronic monoarthritis with positive ACPA and weakly positive RF but she did not fulfill the 2010 ACR/EULAR classification criteria of RA. Although the patient may had had RA, other causes of chronic monoarthritis are not ruled out, including spondyloarthropathies, which are more likely to affect the foot and cause chronic monoarthritis. Nor is it mentioned whether the patient smoked, since smoking can be associated with positive ACPA.]

Reply: Thank you for raising important issues. She did not have the history of smoking, and was not present the symptom of spondyloarthropathies during evaluation. We have added the sentences on line28 and 76-78.

On the comment of [I agree with the published information by Yano K., et al, (2018) that the ankle may be an initial manifestation. But, I consider that all these patients eventually will progress to polyarthritis to make a definite diagnosis of RA. This patient had a DAS 28 that places her in remission since 28-joint count assessments do not include examination of the foot and ankle. Did you order joint ultrasounds of MCP and wrist joints? Although, this study is not included in the 2010 ACR/EULAR criteria either.]

Reply: Thank you for raising important issues. We did not order MRI and joint ultrasounds of her MCP and wrist joints. However, she occasionally experienced proximal interphalangeal joint pain in her fingers, but no swelling was present. We shared the decision making on treatment with MTX. So, she was selected to treat with MTX. We have added the sentences on line 84-88.

On the comment of ['The patient requires a follow-up to know if she develops defined rheumatoid arthritis in the future.]

Reply: Thank you for suitable suggestion. We added the sentences on line150-151.

Reviewer 2 Report

The authors describe a patient with subtalar joint monoarthritis.

Comments

1.      Line 69: As the patient did not fulfill the 2020 ACR / EULAR classification criteria for RA prior to surgery, the authors should present evidence that RA criteria, including radiographs of hands and toes, were fulfilled after the surgery prior to onset of MTX treatment. Otherwise, the subject cannot be considered RA patient and all the definitions should be changed.

Author Response

Replies to Reviewer #2

We have revised the indicated parts of the manuscript according to the comments. Corrections in the newly revised manuscript are underlined.

Please note that your review comments are shown in italic below and our replies in non-italic.

On the comment of [Line 69: As the patient did not fulfill the 2020 ACR / EULAR classification criteria for RA prior to surgery, the authors should present evidence that RA criteria, including radiographs of hands and toes, were fulfilled after the surgery prior to onset of MTX treatment. Otherwise, the subject cannot be considered RA patient and all the definitions should be changed.]

Reply: Thank you for suitable suggestion. We added the data of radiographs of hands and toes as supplementary data. She occasionally experienced proximal interphalangeal joint pain in her fingers, but no swelling was present. We shared the decision making on treatment with MTX. So, she was selected to treat with MTX. We have added the sentences on line 84-88.

Round 2

Reviewer 1 Report

This report can be accepted